# Current Status of Circulating Tumor Cells, Circulating Tumor DNA, and Exosomes in Breast Cancer Liquid Biopsies

**DOI:** 10.3390/ijms21249457

**Published:** 2020-12-11

**Authors:** Marta Tellez-Gabriel, Erik Knutsen, Maria Perander

**Affiliations:** Department of Medical Biology, Faculty of Health Sciences, UiT—The Arctic University of Norway, 9011 Tromsø, Norway; erik.knutsen@uit.no (E.K.); maria.perander@uit.no (M.P.)

**Keywords:** liquid biopsy, breast cancer, prediction, prognosis, exosomes, circulating tumor DNA, circulating tumor cells

## Abstract

Breast cancer is the most common cancer among women worldwide. Although the five-, ten- and fifteen-year survival rates are good for breast cancer patients diagnosed with early-stage disease, some cancers recur many years after completion of primary therapy. Tumor heterogeneity and clonal evolution may lead to distant metastasis and therapy resistance, which are the main causes of breast cancer-associated deaths. In the clinic today, imaging techniques like mammography and tissue biopsies are used to diagnose breast cancer. Even though these methods are important in primary diagnosis, they have limitations when it comes to longitudinal monitoring of residual disease after treatment, disease progression, therapy responses, and disease recurrence. Over the last few years, there has been an increasing interest in the diagnostic, prognostic, and predictive potential of circulating cancer-derived material acquired through liquid biopsies in breast cancer. Thanks to the development of sensitive devices and platforms, a variety of tumor-derived material, including circulating cancer cells (CTCs), circulating DNA (ctDNA), and biomolecules encapsulated in extracellular vesicles, can now be extracted and analyzed from body fluids. Here we will review the most recent studies on breast cancer, demonstrating the clinical potential and utility of CTCs and ctDNA. We will also review literature illustrating the potential of circulating exosomal RNA and proteins as future biomarkers in breast cancer. Finally, we will discuss some of the advantages and limitations of liquid biopsies and the future perspectives of this field in breast cancer management.

## 1. Introduction

Breast cancer is the most common invasive cancer and the leading cause of cancer-related deaths among women [1]. The clinical outcome for breast cancer patients has improved over the past decades due to early diagnosis and development of targeted therapy regimens [2]. In Norway, the five-, ten-, and fifteen-year survival rates for breast cancer patients are 90.7%, 83.6%, and 78.3%, respectively [3]. If organ metastasis has occurred at the time of diagnosis (3.9% of the cases in Norway), the five-year survival rate drops to 29.2% [3]. Even though the survival rates are good for patients diagnosed with early breast cancers, some will experience local or distant recurrence [2]. The time between primary diagnosis and potential relapse varies among different breast cancer subtypes, but particularly hormone receptor (HR) positive cancers have a propensity to recur late (>5 years after initial diagnosis [4,5]. Breast cancer recurrence has even been reported as late as 20 years after primary diagnosis [5].

Breast cancer is a heterogeneous group of neoplasms originating from the epithelial cells lining the milk ducts or lobules [2]. Currently, breast cancer is clinicopathologically classified into four subtypes based on the expression of the HRs estrogen receptor (ER) and progesterone receptor (PR), expression of human epidermal growth factor 2 (HER2), and on the proliferative index Ki-67. These subtypes are luminal A (HR+, HER2−, Ki-67 low), luminal B (HR+, HER2−, Ki-67 high or HR+, HER2+), HER2+ (HR−, HER2+), and triple-negative (HR−, HER2−) (Table 1) [2]. Gene expression profiling has further classified breast cancer into at least five molecular subtypes; luminal A, luminal B, HER-enriched, basal-like, and normal-like [6,7]. As for many other cancers, the TMN Staging system is also used for the classification of breast cancer, where the size of the tumor (T), distant metastases (M), and lymph node status (N) are evaluated. In the clinic, therapy decisions are guided by TMN staging, tumor grade, and receptor expression status. Most breast cancer patients in Norway are diagnosed at an early stage [3]. These early-stage patients will often undergo breast-conserving surgery following radiotherapy (RT) [8]. In addition, for HR+ tumors, endocrine therapy is given for 5–10 years [9]. For more advanced tumors, more complex and intensive systemic adjuvant treatment may be necessary for addition to surgery and RT, including chemotherapy, anti-HER2 therapy, or immune therapy. In addition, for patients with larger primary or locally advanced breast cancer, neoadjuvant therapy is often recommended in order to downstage the disease before surgery. Neoadjuvant and adjuvant therapy may consist of toxic drugs with potentially serious and long-term side effects, sometimes with a profound impact on the patient’s quality of life. Better markers and methods for the identification of patients that do not benefit from adjuvant therapy are needed in order to reduce overtreatment of breast cancer patients.

Breast cancer generally presents a high degree of intra-tumor heterogeneity as well as cancer cell plasticity, which may affect the progression, pathological response to treatment, and recurrence [10,11]. Clonal evolution due to genetic instability, causing accumulation of somatic mutations and gene expression changes may eventually cause resistance to therapy and progression to a more aggressive disease [11,12]. Longitudinal monitoring of disease progression and response to treatment is important for disease control, allowing clinicians to change treatment regimens to prolong survival and improve life quality [13]. Current monitoring methods used in clinical practice include biopsy and imaging (mammography and/or ultrasonography, PET, MRI, and X-ray) [14,15,16,17]. A biopsy is an invasive procedure that only allows the examination of a small region of the tumor tissue, and sometimes the procedure is not feasible in the clinic due to inaccessible tumors or the state of the patient. Furthermore, different imaging techniques rarely provide enough information about tumor characteristics to make treatment decisions. Therefore, isolation of cancer-derived material circulating in the bloodstream or other body fluids through liquid biopsies for subsequent analyses has become an attractive alternative strategy. Biomarkers detected through liquid biopsies can potentially aid in early diagnosis, prognostication, and predict response to a specific therapy. Importantly, liquid biopsies can improve longitudinal disease monitoring to detect potential recurrence. The use of liquid biopsies has some advantages over conventional biopsies, such as the ability to capture the intra-tumor heterogeneity between different regions in the same tumor (spatial heterogeneity), as well as between the primary tumor and metastasis in the same patient (temporal heterogeneity) [18,19]. In addition, preservation methods for tissue biopsies can cause biases, an example being formalin fixation causing high levels of C > T/G > A transitions [20]. Liquid biopsies can overcome both preservation and sampling biases, requires shorter collection and analysis time, and is a less invasive method. Despite this, liquid biopsy is not yet implemented as a standard procedure and is used mainly as a complementary test to tissue biopsy.

There are several types of tumor material that can be assessed by liquid biopsies, such as circulating tumor cells (CTCs), cell-free circulating tumor DNA (ctDNA), and extracellular vesicles (EVs) (Figure 1). CTCs represent intact, viable non-hematological cells with malignant features that can be isolated from a blood sample. CTCs are shed from the primary tumors or metastatic lesions into the bloodstream as single cells or clusters. CtDNA is fragmented cell-free DNA (cfDNA) that is released by apoptotic and necrotic tumor cells. Such DNA fragments may harbor cancer-specific mutations that have occurred in the originating cell. In a similar manner, cancer cell-derived EVs carry cargo that reflects the content of the originating cell. This review aims to describe the most recent results from breast cancer studies analyzing CTCs, ctDNA, and exosomes in liquid biopsies. We will highlight some of the advantages and limitations of liquid biopsies and the future perspectives of this field.

## 2. Liquid Biopsies on Breast Cancer

### 2.1. Circulating Tumor Cells

Circulating tumor cells (CTCs) are cancer cells that have been shed from the primary tumor or a metastatic lesion into the bloodstream. Even though millions of cancer cells in some primary tumors are shed every day, only a sub-fraction of the CTCs might have the propensity to form distant metastases [21,22,23]. Importantly, in breast cancer, both the number and the characteristics of the CTCs can provide critical information about disease progression and response to therapy [24,25,26,27,28,29]. Therefore, great effort has been put forth to develop protocols enabling detection and characterization of CTCs through liquid biopsies. Molecular characterization of CTCs has provided initial evidence demonstrating that CTCs can serve as surrogates for metastatic disease [30]. Moreover, CTCs isolated from breast cancer patients at early stages can be used as a prognostic tool [31]. Finally, CTCs can be used in real-time monitoring therapy responses at different time points during the disease course and for the detection of relapses [32,33]. Although studying CTCs is a promising approach for better characterizing cancers, there are certain issues inherent to the nature of CTCs that should be considered. CTCs are rare events to the point that it is estimated that there will be only one tumor cell per million blood cells in a sample [34]. Thus, it is vital to have a cancer cell-specific enrichment step before analyzing the features of CTCs. Current approaches for isolation and enrichment include a wide range of commercially available technologies based on either biological properties (cell surface protein expression, viability, invasive capacity) or physical properties (size, density, electric charges, deformability) that distinguish cancer cells from surrounding normal hematopoietic cells. These techniques have recently been reviewed [34,35]. Today, the most frequently used strategy for isolating CTCs is based on capturing cells that express the surface molecule epithelial cell adhesion molecule (EpCAM) [36,37]. EpCAM is considered a common marker protein of epithelial-derived cancers and is involved in several cancer-relevant cellular mechanisms like proliferation and cancer cell stemness [38]. Importantly, EpCAM is not expressed in leukocytes [38]. During the past ten years, there has been an increase in the number of methodologies using EpCAM for the detection of CTCs in the blood of carcinoma patients, such as Magsweeper, MACS, Strep-Tactin, CTC-Chip, and the GO Chip [39]. However, the only method for CTC analyses approved by the U.S. Food and Drug Administration (FDA) is the CellSearch^®^ device [40]. The device was implemented on breast cancer already in 2004 in a study showing that the number of CTCs detected before therapy could predict survival in patients with metastatic breast cancer [28]. Briefly, this method is based on the extraction of 7.5 mL blood into special tubes that are centrifuged to separate solid blood components from plasma. Then, by using ferrofluid nanoparticles coated with EpCAM-targeting antibodies, CTCs are magnetically separated from the bulk of other cells present in the blood. CTCs are then stained with monoclonal antibodies towards cytokeratins (CK18 and CK19) that are specific to epithelial cells. Moreover, CD45 staining is used in order to identify possible contaminating leukocytes remaining in the sample. The cartridge containing stained CTCs is scanned to quantify the tumor cells. In this review we will include the most recent clinical studies that have demonstrated the prognostic value of CTC detection and enumeration in early and metastatic breast cancer using the CellSearch^®^ device (Table 2). Previously in 2016, two other reviews have been published on the same topic [30,41].

Several studies have shown that detection of CTCs at the time of diagnosis (baseline) and at different following-up time points during or after completion of therapy can have a prognostic value for early breast cancer patients. This was recently demonstrated by Trapp et al., who conducted a retrospective study, including node-positive or high-risk node-negative primary invasive breast cancer patients from the SUCCESS A trial (NCT02181101) [42]. The main aim of the study was to evaluate CTCs as a prognostic surveillance-marker during routine follow-up of breast cancer after chemotherapy. They found that the presence of CTCs two years after chemotherapy was associated with decreased disease-free as well as overall survival. Moreover, they reported that the CTC status at the two-year follow-up was an independent prognostic factor of the CTC status at baseline [42]. In another study, Sparano et al. evaluated the prognostic capacity of CTC detection five years after primary diagnosis of lymph-node-positive or high-risk lymph-node-negative breast cancer patients included in the E5103 clinical trial (NCT00433511) [43]. They found that detection of ≥1 CTC in blood from HR+ patients at the five-year follow-up was associated with a higher risk of late recurrence (defined as recurrence five years or later after primary diagnosis) [43]. Similarly, Hall et al. assessed the prognostic value of CTCs identified in non-metastatic breast cancer patients before surgical resection [44]. They found that the presence of ≥1 CTCs before resection was negatively associated with relapse-free and overall survival. This was independent of primary tumor size, grade, and lymph node positivity. Pierga et al. assessed the prognostic and predictive potential of CTCs in data from two phase II multicenter trials (BEVERLY-1 and -2, NCT00820547 and NCT00717405), including patients with non-metastatic inflammatory breast cancer, both HER2+ and HER2− [45]. They found that patients with detectable CTCs at baseline had shorter three-year disease-free survival and overall survival as compared to CTC-negative patients. They also reported that patients that both displayed a pathologic complete response and were negative for CTCs at baseline constituted a subgroup of inflammatory breast cancer patients with particularly high overall survival [45]. Recently, a comprehensive meta-analysis of data from 21 different studies of early-stage non-metastatic breast cancer patients with known CTC baseline counts was reported [46]. The authors concluded that baseline CTC detection was an independent prognostic factor for loco-regional relapse-free intervals, distant disease-free survival, and overall survival, which can be added to well-established prognostic models, including clinicopathological data and pathological complete response status [46]. CTC counts can also predict which patients will benefit from additional radiotherapy after surgery. Goodman et al. studied the effect of radiotherapy on survival of early-stage breast cancer patients that were grouped according to CTC detection at baseline [47]. Whereas radiotherapy after breast-conserving surgery was associated with increased survival within the CTC-positive group, no additional effect of radiotherapy was observed for the CTC-negative group. This indicates that CTC counts could be useful in guiding clinicians in determining which patients will benefit from radiotherapy. CTCs have also been shown to improve the prognosis of metastatic breast cancers. For instance, Larsson et al. evaluated whether detected CTCs and CTC clusters both at baseline and after one, three and six months of systemic therapy, was associated with progression-free survival and overall survival [48]. Although ≥5 CTCs and the presence of CTC clusters at baseline indeed were associated with worse prognosis, they found an increased prognostic value of CTC detection over time, suggesting that the dynamic changes in CTC levels are more relevant to prognosis than only baseline enumeration. Moreover, they concluded that the presence of CTC clusters added more significant prognostic value than simple CTC enumeration. Altogether, these results suggest that monitoring CTCs and CTC clusters in blood from metastatic breast cancer patients undergoing systemic therapy may give vital prognostic information about end-point clinical outcomes [48].

As illustrated by the above-mentioned studies, increasing evidence indicates that CTC enumeration can improve prognostication of breast cancer. Other studies have aimed to determine whether receptors on the surface of the CTCs can predict response to therapy. Receptor activator of nuclear factor-kappaB ligand (*RANKL*) and its receptor *RANK* play critical roles in breast cancer development, both during initial tumorigenesis and in the formation of secondary tumors in the bone [49]. Today, breast cancer patients with bone metastases are treated with denosumab, an inhibitory antibody targeting RANKL. Recently, Pantano et al. showed that RANK-expressing CTCs were present in blood from breast cancer patients with skeletal metastases [50]. Moreover, the presence of RANK-positive CTCs was associated with an increased response to denosumab treatment. The authors suggested that monitoring RANK-positive CTCs during denosumab treatment might be an important strategy aiding therapy decisions for bone metastatic breast cancers [50]. In HER2+ breast cancers, several researchers have hypothesized that the presence of HER2+ CTCs can predict the response to HER2− directed therapy. Recently, Jaeger et al. reported that HER2 expression in primary tumors and CTCs isolated from patients after surgery was discordant [51]. They hypothesized that this might have important implications for the use of HER2 targeted therapy. Therefore, they suggested that the HER2 status needs to be reassessed after surgery before initiating adjuvant HER2− targeted therapy [51].

In addition to all the reported clinical trials, there are currently many ongoing studies trying to prove the usefulness of CTCs for breast cancer prediction and prognosis. These studies have been recently reviewed by Schochter et al. [52].

### 2.2. Cell-Free DNA and Circulating Tumor DNA

DNA is released from both healthy and cancer cells into circulation. Cell-free DNA (cfDNA) found in circulation is highly fragmented and most likely originates from apoptotic and necrotic cells, but the exact origin and mechanism of release are still being investigated [58]. Importantly, primary tumors, circulating tumor cells, and occult and overt metastases release DNA at a higher rate than normal cells, and because circulating tumor DNA (ctDNA) displays mutations characteristic of the progenitor tumor, it can serve as a biomarker for diagnosis, prognosis, and prediction [59]. However, the total amount of ctDNA may represent as low as 0.01% of the total cfDNA. These extremely low concentrations make detection challenging, particularly at the early stages of tumor development as tumor burden is low [60]. Despite these difficulties, researchers have worked out several strategies for the detection of ctDNA in breast cancer. There are two fundamental strategies to analyze ctDNA: (1) Quantification of total cfDNA based on the notion that cancer cells release more DNA than healthy cells, and (2) Specific analysis of ctDNA most often based on available mutation data from tumor tissue.

Once total cfDNA is isolated from the blood, it can be measured by fluorescence-based methods or by real-time quantitative PCR (RT-qPCR) [61]. Tangvarasittichai et al. compared the levels of cfDNA in plasma from breast cancer patients and healthy donors, measured by a fluorometer [62]. They reported significantly higher amounts of plasma cfDNA in stage II, III, and IV breast cancer patients at baseline than in healthy controls. Moreover, tumor size, TNM stage, and metastasis were all significantly correlated with the amount of cfDNA [62]. Importantly, after surgery, the cfDNA level dropped, indicating that cfDNA detection can be used to evaluate the efficacy of surgery. Despite the simplicity of the method and that it could easily be incorporated into clinical practice, the diagnostic value of cfDNA quantitation is limited due to the significant overlap of cfDNA concentration in normal and cancer patients and the low ability to distinguish early breast cancer from healthy patients [63].

Measuring the level of ctDNA in cancer patients is a more specific method than measuring cfDNA, and for this reason, it could have a greater potential as a novel method for oncological applications. ctDNA fluctuations during the course of the disease could be correlated with clinical outcome. It was demonstrated by Kruger et al. that high ctDNA molecule numbers and at least two breast cancer-specific mutations, measured by next-generation sequencing (NGS), correlate with poor clinical outcome in ER-positive, HER2-negative metastatic breast cancer patients [64]. Over the last decade, several methodologies have been developed as new viable tools for genomic screening of ctDNA, allowing early detection and management of breast cancer. There are two different approaches for the genomic analysis of ctDNA: (1) Targeted approaches in which a single or few tumor-specific mutations known are used for monitoring residual disease and (2) Untargeted screening that aims at genome-wide analysis for copy number aberrations or point mutations by whole-genome sequencing (WGS) or whole-exome sequencing (WES) [65]. The main disadvantage of targeted strategies is that detailed information about the tumor genome is required. Targeted monitoring can, however, be extremely sensitive, as mutations can be detected at an allele frequency down to 0.01%, with high specificity and at fast and cost-effective rates [66]. Advantages of untargeted strategies include its ability to identify novel changes occurring during tumor treatment and the fact that prior information about the primary tumor’s genome is not required. However, high concentrations of ctDNA are required for reliable reconstruction of tumor-specific genome-wide changes, and usually, they show lower sensitivity (5–10%) [67]. In the following paragraphs different targeted strategies will be reviewed, pointing out those that have been used in breast cancer studies (Table 2).

Detection and quantification of predetermined alleles from ctDNA by Sanger sequencing, single and multi-locus PCRs, and RT-qPCR have established methods in clinical oncology [68,69]. However, these methods are limited to detecting mutant allele frequencies (MAFs) well above 1%, which would leave many cancers smaller than 10 cm^3^ undetected because the expected mutation-carrying ctDNA concentration in plasma for these patients can be 0.1% or lower of total cfDNA [70,71]. Advances in digital PCR, such as digital droplet PCR (ddPCR) and the BEAMing technology, allow absolute quantification of allele frequencies as low as 0.01% [65]. Both technologies have been shown to be able to detect *PIK3CA* mutations in plasma ctDNA from breast cancer patients [53,72]. *PIK3CA* encodes the catalytic p110a subunit of PI3-kinase and is mutated in 30–40% of all breast cancer [73]. By using ddPCR, Kodahl et al. recently showed that in breast cancer patients with advanced disease, there is a strong concordance between *PIK3CA* mutations in metastatic tumor tissue and serum ctDNA [74]. Moreover, although including only a limited number of samples, they observed that the level of *PIK3CA* mutated ctDNA dropped in patients that displayed response to therapy. This suggests that ddPCR-mediated detection of ctDNA *PIK3CA* mutations can be used in early monitoring of therapy response [53]. Several studies have indeed shown that ctDNA *PIK3CA* mutations can be detected in metastatic breast cancer patients [75,76,77]. In contrast, Beaver et al. focused their analyses on blood samples from 15 early-stage breast cancer patients that harbored *PIK3CA* mutations in the primary tumors [54]. In 14 patients (93%), concordant *PIK3CA* mutations were detected in ctDNA in blood samples taken before surgery. CtDNA *PIK3CA* mutational status was also analyzed after surgery for some of the patients, and 5 out of 10 still had detectable ctDNA *PIK3CA* mutations. However, further studies have to be undertaken to assess whether detected *PIK3CA* mutated ctDNA after surgery is an identifier of patients with a higher risk of recurrence [54]. Interestingly, a recent device detecting *PIK3CA* mutations on ctDNA has been approved by the FDA. The therascreen *PIK3CA* RGQ PCR Kit is a laboratory test able to detect 11 mutations in the *PIK3CA* gene from patients with advanced or metastatic breast cancer. The presence of *PIK3CA* mutations is linked with response to treatment with PIQRAY^®^ (alpelisib) [78]. Thus this laboratory test may help doctors identify breast cancer patients who should be treated with PIQRAY^®^. Another attractive target for ctDNA mutational analyses is the *ESR1* gene encoding estrogen receptor alpha (ERα). *ESR1* mutations are associated with resistance to aromatase inhibitors (AIs) that are used as estrogen-deprivation therapy in postmenopausal women [79,80]. Importantly, several studies have shown that the *ESR1* mutations can be selected during AI treatment of metastatic HR+ breast cancers, which can be detected in ctDNA [55,81,82]. In a comprehensive study, Chandarlapaty et al. performed allele-specific ddPCR of *ESR1* mutations Y537S and D538G, which lead to ligand-independent constitutive activation of ERα in blood samples from 541 ER+/HER2− breast cancer patients with metastatic diseases that all had undergone AI therapy in different settings (BOLERO-2, NCT00863655) [55]. They detected *ESR1* Y537S and/or D538G mutations in ctDNA in 28.8% of the patients. Importantly, the presence of the *ESR1* ctDNA mutations was associated with reduced overall survival. Thus, monitoring *ESR1* ctDNA mutations can be important in disease control of metastatic ER+ cancers and potentially give valuable information on which patients will benefit from hormonal therapy.

The above-mentioned studies indicate that digital PCR is a sensitive method that is suited for detecting allele variation in ctDNA from clinical samples. A recent study assessed the agreement between the BEAMing and ddPCR techniques in detecting *PIK3CA* and *ESR1* ctDNA mutations plasma samples from patients with HR+, HER2− metastatic breast cancer enrolled in phase 3 PALOMA-3 trial (NCT01942135) [83]. The authors concluded that the concordance between the techniques is very good, suggesting their sufficient reliability and reproducibility for clinical applications. Of note, these two technologies have been combined in the clinically validated OncoBEAM™ device, which has been tested in several clinical studies, including breast cancer [72,84,85].

The revolution within massively parallel NGS technologies has opened up a new era within personalized medicine. The sensitivity of these methods enables the detection of patient-specific chromosomal structural aberrations, copy number variations, and single nucleotide polymorphisms (SNPs) in both tissue samples and liquid biopsies. Both unbiased WGS and more targeted approaches may be used. Even though proof-of-concept studies have been reported, ctDNA analyses by WGS approaches appear to be somewhat limited by the low abundance of ctDNA fraction in total cfDNA, short ctDNA fragment lengths, as well as by high error rates in the detection of rare allele variants [86,87]. Therefore, targeted approaches are being developed for ctDNA analyses aimed at specifically detecting cancer-specific mutations that have been identified in the primary tumor. Currently, there are several targeted NGS experimental procedures validated in breast cancer patient ctDNA samples. These will be discussed below.

In 2019, the first custom-built NGS-based ctDNA test, Signatera™, was launched (https://www.natera.com/oncology/signatera-advanced-cancer-detection). The aim of this technology is to improve the detection of minimal residual disease after surgery as well as earlier detection of disease recurrence. Signatera™ provides each individual patient with a customized blood test tailored to match the clonal mutations found in the individual’s tumor tissue by WES. Coombes et al. validated the clinical performance of Signatera™ on breast cancer patient samples [56]. They performed upfront WES of tumor tissue from 49 patients and subsequently designed personalized panels of 16 SNPs or indels that were targeted by multiplex sequencing of plasma ctDNA. Importantly, this method was able to detect disease relapse across different subtypes of breast cancer up to two years earlier than imaging in patients with early-stage breast cancer. Other clinical studies in colon cancer [88], lung cancer [89], and bladder cancer [90] support the use of this approach. Recently, it was reported that targeted digital sequencing (TARDIS) of ctDNA might be a highly sensitive method to predict a pathological complete response to neoadjuvant treatment in early and locally advanced breast cancers [57]. Following the WES of tumor biopsies and matched germline samples of 33 patients, the authors analyzed plasma samples for personal mutations (ranging from 6–115 per person) before neoadjuvant treatment. By employing this sensitive strategy, they detected ctDNA in 100% of the samples. They then monitored ctDNA at different time points in 22 patients that were subjected to neoadjuvant treatment and found that ctDNA levels were higher in patients with the residual disease than those that displayed a pathological complete response. The authors argue that this is a promising test to identify women in whom tumors are effectively eliminated by the pre-surgery treatment and, therefore, may not get any additional benefits from undergoing surgery.

ctDNA is believed to be shed across the entire tumor and by all tumor sites, including metastases; thus, it may be a useful tool for addressing tumor heterogeneity. In a proof-of-principle study, De Mattos-Arruda et al. subjected primary tumor, lung metastasis, and plasma samples from a metastatic ER+/HER2− breast cancer patient to massively parallel sequencing targeting 300 genes that are frequently mutated in cancer [91]. All mutations identified in the primary tumor and metastatic lesion were captured in the ctDNA. Importantly, the mutant allele fractions identified in ctDNA varied over time and reflected the clinical response to targeted therapy. Likewise, Murtaza et al. performed a comprehensive case study of an ER+/HER2+ metastatic breast cancer patient with multiple metastatic lesions [91]. They conducted exome and targeted deep amplicon sequencing on samples taken from the primary tumor, several metastatic sites, and serial samples of blood taken throughout the course of the disease. Importantly, ctDNA captured ubiquitous stem mutations (detected in all tumor samples), metastatic-clade mutations (detected in metastatic lesions), and private mutations (only detected in one specific tumor sample). Moreover, disease progression at specific metastatic lesions was reflected in changes in circulating levels of specific private somatic mutations. This indicates that intratumor heterogeneity and clonal evolution can be monitored by serial analyses of ctDNA, which may improve disease control and guide therapy decisions.

### 2.3. Circulating Exosomal RNA Species and Proteins

Hence, far, we have focused our discussion on CTCs and ctDNA as potential biomarkers in breast cancer. These have by far been the most extensively studied analytes in clinical studies due to the implementation of devices like CellSearch^®^, and the recent innovation within technologies for mutational analyses like ddPCR, WES/WGS, and targeted deep sequencing. However, other classes of tumor-derived molecules such as proteins and RNA-species that are released into circulation may potentially provide vital information about cancer development and progression. In this chapter, we will review literature describing studies suggesting that circulating exosomal RNA and proteins detected through liquid biopsies may potentially be implemented as diagnostic, prognostic, or predictive biomarkers (Table 3).

All cells produce and release membranous vesicles into the extracellular environment. Extracellular vesicles (EVs) are generally heterogeneous but can broadly be divided into two classes based on size and mechanisms of biogenesis [92]. These are microvesicles (also referred to as ectosomes) of 100–1000 nm that are formed by plasma membrane shedding, and exosomes of 30–150 nm that derive from the endocytic pathway and are released when multivesicular bodies (MVBs) fuse with the plasma membrane [92,93]. Exosomes, in particular, have gained substantial attention over the last few years as experimental evidence has demonstrated that they contribute to the development and progression of human diseases, including cancer [92,94,95]. It has been demonstrated that cancer cells release more exosomes compared to normal cells as a response to pathophysiological conditions like hypoxia and low pH in the tumor microenvironment or to oncogenic activities within the cells [96,97,98]. Exosomes carry biomolecular cargo, including proteins, lipids, DNA, and RNA species that reflect the content of their originating cells [92,99]. They play key roles in intercellular communication and can deliver their content to neighboring cells in a paracrine fashion. Importantly, exosomes are also detected in body fluids, including blood, suggesting that they can act as mediators of long-distance cellular signaling [99,100,101]. In cancer, exosomes may promote tumor cell migration, invasion, and formation of distant metastatic niches [92,94,98]. They have also been shown to play a role in cancer cell immune evasion [102]. As exosomes are present in body fluids and provide a protective compartment for their biological content, they are promising sources for biomarkers that can be detected through liquid biopsies [99,100,101]. The field still faces challenges in terms of standardization of exosome isolation protocols and lack of proper normalization strategies. This has hampered clinical implementation, and the number of registered clinical studies on cancer exosomes is still scarce. However, several published studies have indeed suggested that analyses of exosomes from body fluids can aid early diagnosis and disease progression monitoring of breast cancer, as well as other cancers [99].

Exosomes isolated from breast cancer patients have been shown to have distinct protein- and RNA contents compared to exosomes derived from healthy donors [103,104]. In this regard, exosomal microRNAs (miRNAs) have been the focus of research in several studies. Many miRNAs are incorporated into exosomes and transferred to other cells where they directly regulate target mRNAs [105]. Compared to larger RNA molecules, miRNAs are remarkably stable, which aid their detection in circulating exosomes. Hannafon et al. performed comprehensive RNA-sequencing-based profiling of exosomal miRNAs derived from cultivated breast cancer cell lines [106]. From their list of exosomal miRNAs, they showed that the levels of miR-1246 and miR-21 were significantly higher in plasma exosomes from breast cancer patients than from healthy donors. Blood samples from triple-negative breast cancer patients have been shown to contain increased levels of exosomal miR-373 compared to other breast cancer subtypes, indicating that it is a potential biomarker for diagnosis of basal-like breast cancers [107]. Stevic et al. analyzed the presence of a panel of 45 exosomal miRNAs in plasma from 435 patients diagnosed with either HER2+ or triple-negative breast cancer and indeed confirmed that the exosomal miRNA content differed among the two breast cancer subtypes [108]. Moreover, they reported that specific exosomal miRNAs are associated with clinicopathological parameters, including tumor size, lymph node status, and pathological complete response to neoadjuvant therapy. Recently, it was shown that circulating exosomal miR-21 levels are significantly higher in breast cancer patients with distant metastases compared to those that only had local disease [109]. In HER2+ cancers, the level of miR-21 dropped after treatment with trastuzumab in a neoadjuvant setting, indicating that miR-21 levels could be used in monitoring the treatment response of HER2+ cancers [109]. Sueta et al. compared the expression of 384 exosomal miRNAs in serum from breast cancer patients with recurrence and patients without recurrence [110]. They identified 11 miRNAs that were differentially expressed in the two groups of patients, suggesting that they could be used as prognostic biomarkers. Exosomal miRNAs can also distinguish patients with invasive ductal carcinoma (IDC) from patients diagnosed with ductal carcinoma in situ (DCIS). This is exemplified by miR-223-3p levels that were significantly higher in patients with IDC compared to patients with DCIS or healthy individuals [111]. Importantly, miR-223-3p levels could also predict which DCIS samples ultimately were upgraded to IDC at final pathologic diagnosis. All the above-mentioned studies have been done in exosomes isolated from blood samples. Recently, however, Hirschfeld and coworkers analyzed the expression of 13 miRNAs from exosomes isolated from urine samples from breast cancer patients and healthy volunteers [112]. They suggested that a panel of four urine exosomal miRNAs, including miR-424, miR-423, miR-660, and let7-i, could be used as biomarkers for breast cancer detection.

Even though the majority of studies on circulating exosomal RNA molecules have been focused on miRNAs, a few studies have shown that exosomal mRNAs and long noncoding RNAs (lncRNAs) might be used as biomarkers. Rodriguez et al. showed that a set of metastasis- and stemness-related mRNAs were higher expressed in plasma exosomes from breast cancer patients with poor disease outcomes than in those that were isolated from patients with good outcomes [113]. Increased circulating exosomal mRNA expression of the drug metabolic enzyme glutathione S-transferase P1 (GSTP1) was associated with drug resistance in breast cancer patients [114]. In HR+ breast cancer, elevated circulating exosomal levels of mRNAs encoding cell cycle-regulated thymidine kinase 1 (TK1) and cyclin-dependent kinase 9 (CDK9) were associated with poor clinical response to the CDK4/CDK6 inhibitor palbociclib [115]. Recently, it was reported that higher levels of the lncRNA *HOTAIR* were found in circulating exosomes in breast cancer patients than in healthy individuals [116]. Exosomal *HOTAIR* levels were also associated with poor prognosis, and it was suggested as a possible predictor of response to chemotherapy and tamoxifen treatment. Zhong et al. reported that serum exosomal levels of the lncRNA *H19* were significantly higher in breast cancer patients than healthy controls [117]. Finally, in HER2+ breast cancers, elevated serum exosomal levels of the lncRNA *SNHG14* were associated with resistance to trastuzumab [118].

The protein content of exosomes reflects their mode of biosynthesis, and they are typically enriched with endosomal/lysosomal proteins like tetraspanins (CD9, CD63, CD81, and CD82), ESCRT complex proteins (e.g., ALIX and TSG101), Rab proteins, Syndecan-1, and Syntenin-1 [92,93]. Quantification of exosomal CD82 in serum has indeed been suggested as a strategy for early diagnosis of breast cancer [119]. Heat shock proteins (Hsps), including Hsp90 and Hsp70, are also frequently found in exosomes [120]. Hsps are often upregulated in cancer due to proteotoxic stress [121], and recently, a prospective clinical pilot study reported that plasma from both breast cancer and non-small cell lung cancer patients contained increased levels of exosomal Hsp70 compared to healthy donors [122]. Furthermore, exosomal Hsp70 levels were significantly higher in patients with metastatic disease compared to those with non-metastatic disease. Finally, plasma exosomal Hsp70 levels could predict the response to therapy. Breast cancer patients have also been shown to have elevated levels of circulating exosomal fibronectin, developmental endothelial locus-1 (Del-1), and the 2B splice variant of survivin (survivin-2B), compared to healthy controls [123,124,125,126]. In a study including 32 breast cancer samples, 75% of the patients were found to have increased levels of circulating exosomal glypican-1 (GPC1), a heparan sulfate proteoglycan that is overexpressed in breast cancer, compared to a control group [127]. Exosomes isolated from pleural effusions from breast cancer patients have increased levels of the epithelial marker proteins EpCAM and CD24 compared to controls [128]. Whereas CD24 also was detected in serum exosomes of breast cancer patients, conflicting results have been reported for exosomal EpCAM in blood. Whereas Rupp et al. failed to detect EpCAM on the surface of circulating exosomes in serum possibly due to its proteolytic cleavage by serum proteases, Fang and coworkers were able to isolate and detect increased levels of EpCAM-positive exosomes from plasma from breast cancer patients, using a microfluidic chip device [102,128,129]. A comprehensive phosphoproteomic study of plasma exosomes indicated that breast cancer could be diagnosed based on cancer-specific phosphorylation events that are reflected in the phosphoprotein content of circulating exosomes [130]. Three exosomal phosphoproteins, RALGAPA2, PRKG1, and TJP2, were validated and suggested as potential breast cancer biomarkers.

Several reports have shown that the proteome of breast cancer-derived exosomes reflects the subtype of the malignant cells. For instance, it has been shown that ectopic overexpression of the HER2− encoding *ERBB2* gene in the immortalized human mammary luminal epithelial cell line HB4a altered the proteomic landscape of the secreted exosomes [131]. Moreover, the overexpressed HER2 receptor was packed into and could be detected in exosomes. Consistency between the HER2 expression status in the breast cancer tissue and circulating exosomes have been demonstrated [129,132,133]. Finally, Rontogianni et al. showed that triple-negative breast cancers could be distinguished from HER2+ breast cancers by proteomic profiling of circulating exosomes [133].

The protein content of circulating exosomes in breast cancer patients can potentially predict the response to therapy. It has been suggested that the HER2 receptor on exosomes might act as a decoy for trastuzumab, a HER2-targeting monoclonal antibody used in the clinic today [132]. By doing this, exosomal HER2 may directly counteract the efficacy of trastuzumab. Ning et al. showed that circulating exosomal levels of Ubiquitin carboxyl-terminal hydrolase-L1 (UCH-L1), which may upregulate drug resistance-associated P-glycoprotein (P-gp) levels, can predict the response to chemotherapy in breast cancer patients [134]. Similar observations were done for another P-gp regulator, Transient receptor potential channel 5 (TrpC5), that were detected in circulating exosomes from patients that had undergone chemotherapy and found it to be a predictor of acquired chemoresistance in metastatic breast cancer [135,136].

## 3. Discussion and Future Perspectives

In order to have clinical utility, cancer biomarkers should fulfill a set of analytical and clinical criteria [137,138]. High analytical performance of a biomarker, measured as specificity, precision, accuracy, sensitivity, and good linear range, does not necessarily directly imply its clinical validity or utility. The clinical validity of a biomarker is the ability of the biomarker to accurately identify patients with the targeted pathological state, while the utility measures the benefit (such as reduced mortality) of using the biomarker in clinical settings [137,138]. During the last few years, there has been a tremendous progression in the development of sensitive technologies allowing the detection and analyses of tumor materials in body fluids. Cancer-derived cells and molecules can be isolated through liquid biopsies, and several studies have demonstrated that they can give vital information about the diagnosis, disease progression, and therapy response in breast cancer as well as other cancers. Although being promising, the implementation of such biomarkers into the clinical practice requires a better understanding of underlying shedding/releasing mechanisms, as well as dynamics and kinetics of the circulating tumor material. Moreover, there are still important technical issues that need to be resolved to improve reproducibility [139]. Therefore, in spite of the huge effort put in by the scientific community, the majority of liquid biopsy assays still lack evidence of clinical validity and, in particular, clinical utility. Therefore, their use is still mostly confined to research purposes and clinical studies [140].

CTCs might give important information about tumor burden and might also contain molecules reflecting cancer-specific features. An important limitation of CTCs is that they may not fully reflect the biology of the underlying tumor. In addition, cancer cells are not confined to one phenotype and might have epithelial, mesenchymal, or stem-like characteristics [141]. Currently, the CellSearch^®^ platform is the only one licensed by the FDA for the isolation of CTCs and their prognostic enumeration in breast cancer [40]. However, the standard CellSearch^®^ platform enriches CTCs based on the expression of the epithelial marker protein EpCAM. In breast cancer, as well as other cancers, several lines of evidence suggest that the most aggressive tumor cells are those that have undergone epithelial-mesenchymal transition (EMT), a process that is associated with downregulation of EpCAM as well as other adhesion molecules [142,143]. EMT is associated with increased propensity to metastasize, acquired stem cell-like features, and drug resistance [144]. Thus, only focusing on EpCAM-expressing cells might confer serious limitations to the studies, leaving groups of CTCs undetected [145]. Another concern is that some studies use a cutoff of only 1 CTC per 7.5 mL of blood to score a sample as CTC-positive. This indicates that the assay might be prone to inter-laboratory variations in image interpretation, which might impact its robustness. Moreover, rare EpCAM-positive “CTC-like” cells can be detected in healthy women, which might preclude the interpretation of the test [146]. Finally, different types of tumors have different propensity to shed cells. As breast tumors generally display a high degree of heterogeneity, certain subpopulations of cells might more easily be shed into the bloodstream than others [147]. This means that the CTCs detected in liquid biopsies might not reflect the actual intratumoral heterogeneity of a breast tumor. In spite of this, several studies have shown that CTCs reflect the tumor burden, and CellSearch^®^-based CTC detection both at baseline and during therapy might give vital information about residual disease and recurrence. However, further standardization of the assays addressing all the above-mentioned limitations need to be addressed before CTC enumeration can implemented in the clinic as an independent assay to guide treatment decisions [88,89].

Detection of ctDNA that reflects the mutational status and clonal evolution of a tumor holds great promise for future monitoring of cancer progression and drug response in breast cancer. The development of high-sensitivity technologies enabling detection of mutant allele frequencies of <0.01% has, without doubt, revolutionized this field [38,64]. However, ctDNA constitutes only a minor fraction of total cfDNA, and reliable and reproducible detection of low allele frequency variants is still challenging [148]. This might, for instance, hamper the detection of metastatic site-specific private mutations that have occurred during clonal evolution. Moreover, it is important to distinguish ctDNA mutations from randomly mutated cfDNA derived from noncancerous blood cells [147]. Another obvious point is that, as for CTCs, not all tumors shed DNA [147]. As mentioned above, a good concordance was demonstrated between ddPCR and BEAMing techniques in detecting *PIK3CA* and *ESR1* ctDNA mutations in plasma samples from breast cancer patients [83]. In contrast, a recent study reported a significant discordance between four commercial NGS-based ctDNA tests, particularly in the detection of allele frequency variants <1% [149]. Therefore, before the implementation of ctDNA tests in the clinic as independent diagnostic, prognostic, or predictive assays, standardization in terms of how the samples are obtained and how the analyses are performed are needed. However, the prospect of using ctDNA analyses for early detection of disease recurrence is promising and might complement conventional biopsies.

It is generally accepted that secreted exosomes provide a protective compartment for circulating biomolecules. Furthermore, in breast cancer and other cancers, tumor-derived exosomes may contain cargo that can give vital information about disease development and progression. However, as illustrated by the limited number of comprehensive clinical studies on exosomes and breast cancer, the field faces challenges that need to be resolved before being clinically applicable. First of all, further research should be conducted on the selection and packaging mechanisms of cargo in the originating cancer cell to better evaluate whether the exosome content in a given sample is based on specific selection or random incorporation of the secreted cargo. In the latter case, inter- and intraindividual heterogeneity of exosomal content might preclude their implementation as reliable biomarkers. Moreover, sensitive methods for purification of exosomes that excludes contaminants like lipoprotein particles, microvesicles, and protein complexes, are needed [150]. Such methods should be optimized based on the nature of the biomolecules that are going to be analyzed. Multiparametric analyses of different exosomal biomolecules might thus require several samples. Finally, the protocols need to be fast and convenient to be used in clinical practice. Today, the gold-standard method for exosome isolation is ultracentrifugation, which is a tedious process that might require large sample volumes. Nevertheless, with efforts being put in developing new isolation protocols like those based on microfluidics and establishing standards applied to all steps in sampling, purification, and subsequent analyses, circulating exosomal-derived biomarkers might provide additive information refining diagnosis and aiding disease control in breast cancer [151].

Most studies on liquid biopsies have been focused on a single analyte (CTCs, ctDNA, exosomes, etc.). This is about to change, and more initiatives are now being taken to develop multiparametric assays that incorporate data from separate platforms analyzing different analytes. Such combinatorial strategies require powerful statistical tools and algorithms that can integrate and standardize data obtained from different platforms. In this regard, PROLIPSY (NCT04556916) is a multinational effort, which is exploring whether a combination of CTCs, ctDNA, and tumor-secreted exosomes might assist in early detection and prognostic profiling of prostate cancer. CancerSEEK is an already commercialized blood-based assay that looks at ctDNA and protein biomarkers in parallel. Cohen et al. conducted a multicenter study, enrolling 1000 patients bearing different types of cancer. Their results showed that they could achieve early-detection rates of 69–98% for five different malignancies (ovary, liver, stomach, pancreas, and esophagus cancer), with a false-positive rate of less than 1%. Regarding breast cancer, this test could only detect around 30% of the cases, so further improvements are needed [74]. Finally, another example is the study conducted by Ye, Z. et al., where they investigate individual and joint effects of CTC and cfDNA on clinical outcomes of metastatic breast cancer (MBC) patients [152]. The main conclusion was that CTC and total cfDNA levels were individually and jointly associated with progression-free survival and overall survival in metastatic breast cancer patients [152].

## 4. Conclusions

Currently, exhaustive validation in clinical trials is required to boost the clinical utility of liquid biopsies. More studies are needed to improve the test’s accuracy and sensitivity of the tests. Moreover, additional investigations are required to clarify whether liquid biopsy provides a representative sampling of all genetic clones within a tumor or if there is a bias to specific subregions of the tumor. Breast cancer is a heterogeneous disease, but the survival rate is good if the patient is diagnosed before distant metastasis has occurred. In many countries, mammography screening programs are implemented for women above a certain age (50 years in Norway), which indeed has led to the early detection and diagnosis of many breast cancers. Early detection might potentially also lead to overtreatment of some patients. In addition, current diagnostic tools for breast cancer are well-established. Based on this, it is unclear whether liquid biopsies can add additional value in breast cancer diagnosis. In sharp contrast, longitudinal monitoring tools that can detect residual disease after therapy, inform about disease progression and predict cancer recurrence at an early stage are desperately needed. Here, biomarkers acquired through liquid biopsies can make a tremendous impact on the future control of breast cancer.

## Figures and Tables

**Figure 1 ijms-21-09457-f001:**
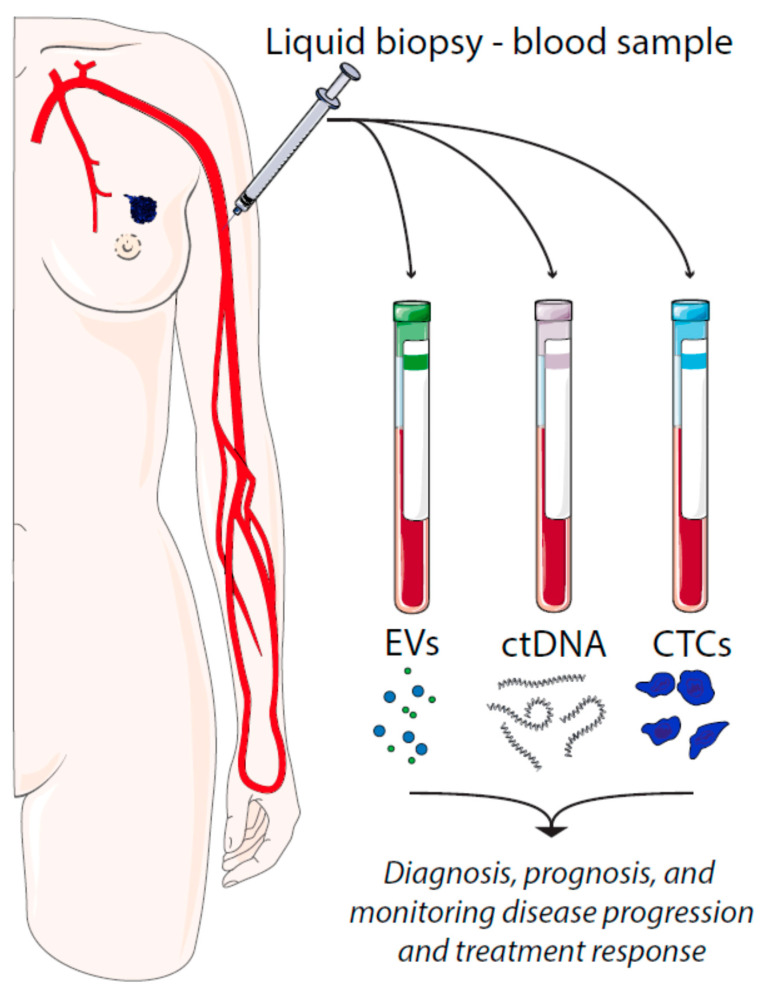
Liquid biopsies in breast cancer and their utilities. EV = extracellular vesicles, ctDNA = circulating tumor DNA, CTCs = circulating tumor cells.

**Table 1 ijms-21-09457-t001:** Clinicopathological classification of breast cancer subtypes.

Subtype	HR Status	HER2 Status	Ki-67 Expression
Luminal A	Positive	Negative	Low
Luminal B	Positive	Negative	High
Luminal B	Positive	Positive	-
HER2+	Negative	Positive	-
Triple-negative	Negative	Negative	-

HR = hormone receptor include both estrogen receptor (ER) and progesterone receptor (PR), HER2 = human epidermal growth factor 2, Ki-67 = proliferative index.

**Table 2 ijms-21-09457-t002:** Examples of studies demonstrating the clinical relevance of circulating tumor cells and ctDNA.

Early Breast Cancer/Advanced Disease	Liquid Biopsy	Method	Prognostic/Predictive Value or Monitoring	References
Early breast cancer	CTCs	CellSearch^®^	Prognostic	[42,44,45,46]
Advanced disease	CTCs	CellSearch^®^	Prognostic	[43]
Advanced disease	CTCs	CellSearch^®^	Prediction	[47,50]
Early breast cancer	CTCs	CellSearch^®^	Prediction	[48,51]
Advance disease	ctDNA	ddPCR	Prediction and monitoring	[53]
Early breast cancer	ctDNA	ddPCR	Prognostic	[54]
Advanced disease	ctDNA	ddPCR	Prediction and prognostic	[55]
Advanced disease	ctDNA	Signatera™	Prognostic	[56]
Early breast cancer and Locally advanced disease	ctDNA	TARDIS	Prediction and monitoring	[57]

**Table 3 ijms-21-09457-t003:** Examples of studies suggesting the potential of circulating exosomal RNA and proteins as liquid biopsies in breast cancer.

Exosomal Content	Prognostic/Predictive Value/Monitoring/Diagnostic	References
miRNA	Diagnosis	[106,107,108,109,110,111,112]
miRNA	Prognosis	[109,110]
miRNA	Prediction	[108]
miRNA	Monitoring	[109]
mRNA/lncRNA	Diagnosis	[116,117]
mRNA/lncRNA	Prognosis	[109,114]
mRNA/lncRNA	Prediction	[118]
Protein	Diagnosis	[119,122,123,124,125,126,127,130]
Protein	Prediction	[134,135,136]

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
