# Peer review of "Current Status of Circulating Tumor Cells, Circulating Tumor DNA, and Exosomes in Breast Cancer Liquid Biopsies"

_ijms, 2020, doi:10.3390/ijms21249457_

Round 1
Reviewer 1 Report
A very interesting manuscript that covers the whole problem of circulating tumor cells, circulating tumor DNA, and exosomes in breast cancer liquid biopsies.The authors undertook a comprehensive discussion of the topic and did it very clearly. The authors cited most of the current literature and looked at them critically.
Author Response
Dear reviewer,
Thank you for your consideration of our manuscript to be published. We have reviewed the comments of the reviewers and have revised the manuscript. Please see above in detail the modifications included:
- line 146-147: This was recently demonstrated by Trapp et al. who conducted a retrospective study including node positive or high-risk node negative primary invasive breast cancer patients from the SUCCESS A trial (NCT02181101).
- line 160-162: Pierga et al. assessed the prognostic and predictive potential of CTCs in data from two phase II multicenter trials (BEVERLY-1 and -2, NCT00820547 and NCT00717405) including patients with non-metastatic inflammatory breast cancer, both HER2+ and HER2-.
- line 238-240: It was demonstrated by Kruger et al. that high ctDNA molecule numbers and at least two breast cancer specific mutations, measured by next-generation sequencing (NGS), correlate with poor clinical outcome in ER‐positive, HER2‐negative metastatic breast cancer patients.
- line 298-299: plasma samples from patients with HR+, HER2 negative metastatic breast cancer enrolled in the phase 3 PALOMA-3 trial (NCT01942135).
We have attached the new document including the above indicated changes.
Sincerely,

Reviewer 2 Report
Tellez-Gabriel et al have provided a comprehensive overview of clinical studies in breast cancer utilizing liquid biopsies (CTC, ctDNA or EV) to assess clinical potential and utility. They end with a nice discussion, clearly pointing out the pro’s and con’s of each type of liquid biopsies, and what’s required for clinical practice.
I find the review very nicely written; it’s clear, it addresses an important and timely topic which has, to my knowledge, not been reviewed in this form in another journal. As such, I would advise to publish this paper after addressing some minor comments:
- The authors need to check their text for details like double spaces, ) missing, spaces before a . , etc.
- I think it would be valuable to include the breast cancer subtypes that were studied in the clinical trials described. It was done for section 2.3, but not for section 2.1 and 2.2.
Author Response
Dear reviewer,
First, we really appreciate your comments.
We have checked the details, such as double spaces,) missing, spaces before a. and corrected whenever it was necessary. We are sorry for these mistakes.
Regarding the information of the patients included in the studies in the sections 2.1 and 2.2. We modified and wrote as much information as we found. However, for some of the studies we couldn’t find more information than the one that was previously included. Please see below the modifications we have added:
- line 146-147: This was recently demonstrated by Trapp et al. who conducted a retrospective study including node positive or high-risk node negative primary invasive breast cancer patients from the SUCCESS A trial (NCT02181101).
- line 160-162: Pierga et al. assessed the prognostic and predictive potential of CTCs in data from two phase II multicenter trials (BEVERLY-1 and -2, NCT00820547 and NCT00717405) including patients with non-metastatic inflammatory breast cancer, both HER2+ and HER2-.
- line 238-240: It was demonstrated by Kruger et al. that high ctDNA molecule numbers and at least two breast cancer specific mutations, measured by next-generation sequencing (NGS), correlate with poor clinical outcome in ER‐positive, HER2‐negative metastatic breast cancer patients.
- line 298-299: plasma samples from patients with HR+, HER2 negative metastatic breast cancer enrolled in the phase 3 PALOMA-3 trial (NCT01942135).
We have attached the new document including the above indicated changes.
